# HIPPO signaling resolves embryonic cell fate conflicts during establishment of pluripotency in vivo

Tristan Frum[1], Tayler M Murphy[2,3], Amy Ralston[1,2,3]*

[1]Department of Biochemistry and Molecular Biology, Michigan State University, Michigan, United States; [2]Genetics Graduate Program, Michigan State University, Michigan, United States; [3]Reproductive and Developmental Biology Training Program, Michigan State University, Michigan, United States

**Abstract** During mammalian development, the challenge for the embryo is to override intrinsic cellular plasticity to drive cells to distinct fates. Here, we unveil novel roles for the HIPPO signaling pathway in controlling cell positioning and expression of *Sox2*, the first marker of pluripotency in the mouse early embryo. We show that maternal and zygotic YAP1 and WWTR1 repress *Sox2* while promoting expression of the trophectoderm gene *Cdx2* in parallel. Yet, *Sox2* is more sensitive than *Cdx2* to *Yap1/Wwtr1* dosage, leading cells to a state of conflicted cell fate when YAP1/WWTR1 activity is moderate. Remarkably, HIPPO signaling activity resolves conflicted cell fate by repositioning cells to the interior of the embryo, independent of its role in regulating *Sox2* expression. Rather, HIPPO antagonizes apical localization of Par complex components PARD6B and aPKC. Thus, negative feedback between HIPPO and Par complex components ensure robust lineage segregation.
DOI: https://doi.org/10.7554/eLife.42298.001

**\*For correspondence:**
aralston@msu.edu

**Competing interests:** The authors declare that no competing interests exist.

## Introduction

During embryogenesis cells gradually differentiate, adopting distinct gene expression profiles and fates. In mammals, the first cellular differentiation is the segregation of trophectoderm and inner cell mass. The trophectoderm, which comprises the polarized outer surface of the blastocyst, will mainly produce cells of the placenta, while the inner cell mass will produce pluripotent cells, which are progenitors of both fetus and embryonic stem cells. Understanding how pluripotent inner cell mass cells are segregated from non-pluripotent cells therefore reveals how pluripotency is induced in a naturally occurring setting.

Progenitors of inner cell mass are first morphologically apparent at the 16 cell stage as unpolarized cells residing inside the morula (reviewed in *Frum and Ralston, 2017*). However, at this stage, pluripotency genes such as *Pou5f1* (*Oct4*) and *Nanog*, do not specifically label inside cells (*Dietrich and Hiiragi, 2007*; *Niwa et al., 2005*; *Palmieri et al., 1994*; *Strumpf et al., 2005*). Thus, the first cell fate decision has been studied mainly from the perspective of trophectoderm specification because the transcription factor CDX2, which is essential for trophectoderm development (*Strumpf et al., 2005*), is expressed specifically in outer cells of the 16 cell embryo (*Ralston and Rossant, 2008*), and has provided a way to distinguish future trophectoderm cells from non-trophectoderm cells. Knowledge of CDX2 as a marker of trophectoderm cell fate enabled the discovery of mechanisms that sense cellular differences in polarity and position in the embryo, and then respond by regulating expression of *Cdx2* (*Nishioka et al., 2009*). However, the exclusive study of *Cdx2* regulation does not provide direct knowledge of how pluripotency is established because the absence

**eLife digest** As an embryo develops, its cells divide, grow and migrate in specific patterns to build an organized collection of cells that go on to form our tissues and organs. One of the first steps – well before the embryo has implanted into the womb – is to allocate cells to make part of the placenta.

Once this process is complete, the remaining cells continue building the organism. These cells are pluripotent, meaning they can develop into any part of the body. Scientists think that the embryo manages to sort 'placenta cells' from pluripotent ones with the help of certain proteins, which the mother has packaged into her eggs.

To investigate this further, Frum et al. used genetic tools to track a specific gene called Sox2 that identifies pluripotent cells as soon as they are formed in mouse embryos. The experiments revealed that the mother places two closely related proteins known as YAP1 and WWTR1 within each egg, which help to make placenta cells different from pluripotent cells. Moreover, both proteins enable the embryo to segregate these two cell types to two different locations: placenta cells are moved to the outer layer of the embryo, while pluripotent cells are moved to the inside.

Current technologies allow researchers to create pluripotent cells in the laboratory. But these approaches often result in error, failing to replicate the embryo's natural ability. By studying how embryos form and arrange pluripotent cells, scientists hope to advance stem cell technology (which emerge from pluripotent cells). This may help to find new ways to heal damaged tissues and organs, or to treat or even prevent many diseases.

DOI: https://doi.org/10.7554/eLife.42298.002

of *Cdx2* expression does not necessarily indicate acquisition of pluripotency. As such, our understanding of the first cell fate decision in the early mouse embryo is incomplete.

In contrast to other markers of pluripotency, *Sox2* is expressed specifically in inside cells at the 16 cell stage, and is therefore the first marker of pluripotency in the embryo (*Guo et al., 2010*; *Wicklow et al., 2014*). The discovery of how *Sox2* expression is regulated in the embryo therefore provides unique insight into how pluripotency is first established in vivo. Genes promoting expression of *Sox2* in the embryo have been described (*Cui et al., 2016*; *Wallingford et al., 2017*). However, it is currently unclear how expression of *Sox2* becomes restricted to inside cells. We previously showed that *Sox2* is restricted to inside cells by a *Cdx2*-independent mechanism (*Wicklow et al., 2014*), which differs from *Oct4* and *Nanog*, which are restricted to the inner cell mass by CDX2 (*Niwa et al., 2005*; *Strumpf et al., 2005*). Thus, *Sox2* and *Cdx2* are regulated in parallel, leading to complementary inside/outside expression patterns. However, it is not known whether *Sox2* is regulated by the same pathway that regulates *Cdx2* or whether a distinct pathway could be in use.

The expression of *Cdx2* is regulated by members of the HIPPO signaling pathway. In particular, the HIPPO pathway kinases LATS1/2 become active in unpolarized cells located deep inside the embryo, where they antagonize activity of the YAP1/WWTR1/TEAD4 transcriptional complex that is thought to promote expression of *Cdx2* (*Anani et al., 2014*; *Cockburn et al., 2013*; *Hirate et al., 2013*; *Kono et al., 2014*; *Korotkevich et al., 2017*; *Leung and Zernicka-Goetz, 2013*; *Lorthongpanich et al., 2013*; *Mihajlović and Bruce, 2016*; *Nishioka et al., 2009*; *Nishioka et al., 2008*; *Posfai et al., 2017*; *Rayon et al., 2014*; *Watanabe et al., 2017*; *Yagi et al., 2007*; *Zhu et al., 2017*). In this way, the initially ubiquitous expression of *Cdx2* becomes restricted to outer trophectoderm cells. However, the specific requirements for *Yap1* and *Wwtr1* in the regulation of *Cdx2* has been inferred from overexpression of wild type and dominant-negative variants, neither of which provide the standard of gene expression analysis that null alleles can provide. Nonetheless, the roles of *Yap1* and *Wwtr1* in regulating expression of *Sox2* have not been investigated. Here, we evaluate the roles of maternal and zygotic YAP1/WWTR1 in regulating expression of *Sox2* and cell fate during blastocyst formation.

## Results

### Patterning of *Sox2* is ROCK-dependent

To identify the mechanisms regulating *Sox2* expression during blastocyst formation, we focused on how *Sox2* expression is normally repressed in the trophectoderm to achieve inside cell-specific expression. We previously showed that SOX2 is specific to inside cells in the absence of the trophectoderm factor CDX2 (*Wicklow et al., 2014*), suggesting that mechanisms that repress *Sox2* in the trophectoderm act upstream of *Cdx2.* Rho-associated, coiled-coil containing protein kinases (ROCK1 and 2) are thought to act upstream of *Cdx2* because embryos developing in the presence of a ROCK-inhibitor (Y-27632, ROCKi) exhibit reduced *Cdx2* expression (*Kono et al., 2014*). Additionally, quantitative RT-PCR showed that *Sox2* mRNA levels are elevated in ROCKi-treated embryos (*Kono et al., 2014*), suggesting that ROCK1/2 activity leads to transcriptional repression of *Sox2*. However, the role of ROCK1/2 in regulating the spatial expression of *Sox2* has not been investigated.

To evaluate the roles of ROCK1/2 in patterning *Sox2* expression, we collected 8-cell stage embryos prior to embryo compaction (E2.5), and then cultured these either in control medium or in the presence of ROCKi for 24 hr (*Figure 1A*). Embryos cultured in control medium exhibited normal cell polarity, evidenced by the apical localization of PARD6B and basolateral localization of E-cadherin (CDH1) in outside cells (*Figure 1B,C*) as expected (*Vestweber et al., 1987*; *Vinot et al., 2005*). Additionally, SOX2 was detected only in inside cells in control embryos (*Figure 1C,D*). By contrast, embryos cultured in ROCKi exhibited defects in cell polarity (*Figure 1B', C'*), consistent with prior studies (*Kono et al., 2014*). Interestingly, in ROCKi-treated embryos, we observed ectopic SOX2 expression in cells located on the outer surface of the embryo (*Figure 1C', D*), indicating that ROCK1/2 participates in repressing expression of *Sox2* in the trophectoderm.

To scrutinize the identity of outside-positioned SOX2-positive cells in ROCKi-treated embryos, we co-stained an additional cohort of control and ROCKi-treated embryos with CDX2 and SOX2 and compared the overlap of lineage marker expression. In control embryos, CDX2 was detected only in outside cells (*Figure 1—figure supplement 1A*) as expected at this stage (*Ralston and Rossant, 2008*; *Strumpf et al., 2005*). In ROCKi-treated embryos, CDX2 expression levels were reduced (*Figure 1—figure supplement 1A'*) as was the proportion of outside cells in which CDX2 was detected (*Figure 1E*), as previously reported (*Kono et al., 2014*). However, among outside cells, a substantial proportion coexpressed CDX2 and SOX2 in ROCKi-treated embryos compared with controls (*Figure 1E* and *Figure 1—figure supplement 1A*), suggesting that ROCK inhibition leads to an increase in outside cells of mixed lineage. Since SOX2 expression does not regulate expression of CDX2 (*Wicklow et al., 2014*), these observations suggest that ROCK1/2 activity regulates these genes through parallel mechanisms. We next sought to identify mediators that act downstream of ROCK1/2 to repress expression of *Sox2* in the trophectoderm.

### YAP1 is sufficient to repress expression of SOX2 in the inner cell mass

Several direct and indirect targets of ROCK1/2 kinases in the early embryo have been described (*Alarcon and Marikawa, 2018*; *Shi et al., 2017*). Among these is YAP1, a transcriptional partner of TEAD4 (*Nishioka et al., 2009*), since ROCK activity is required for the nuclear localization of YAP1 (*Kono et al., 2014*). Notably, *Tead4* is required to repress expression of *Sox2* in the trophectoderm (*Wicklow et al., 2014*), consistent with the possibility that YAP1 partners with TEAD4 to repress *Sox2* expression in the trophectoderm. To test this hypothesis, we overexpressed a constitutively active variant of YAP1 (YAP1$^{CA}$). Substitution of alanine at serine 112 leads YAP1 to be constitutively nuclear and constitutively active (YAP1$^{CA}$ hereafter) (*Dong et al., 2007*; *Nishioka et al., 2009*; *Zhao et al., 2007*). We injected mRNAs encoding YAP1$^{CA}$ and GFP into one of two blastomeres at the 2-cell stage, and then cultured these to the blastocyst stage (*Figure 1F*). This mosaic approach to overexpression permitted comparison of *Yap1$^{CA}$*-overexpressing with non-injected cells, which served as internal negative controls. We first examined localization of YAP1 in these embryos at the morula stage, with the expectation that YAP1 would be detected in nuclei of both inside and outside cells in *YAP1$^{CA}$*-overexpressing cells (*Nishioka et al., 2009*). As expected, YAP1 was observed in nuclei of all *Yap1$^{CA}$*-overexpressing cells (*Figure 1—figure supplement 1B,C*). We next evaluated the consequences of ectopic nuclear YAP1 on expression of SOX2 in inside cells. We observed a

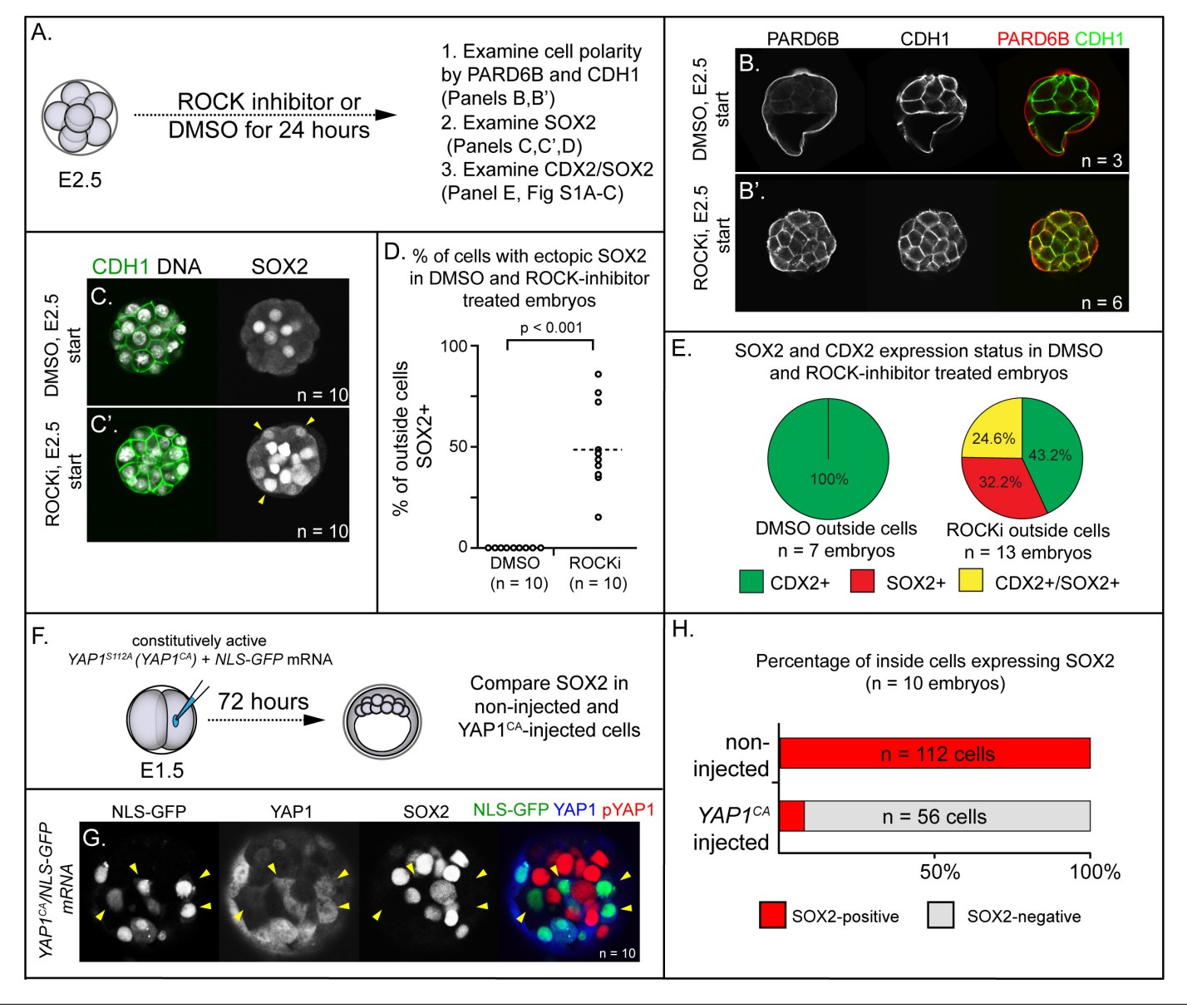

**Figure 1.** ROCK1/2 and nuclear YAP1 repress expression of SOX2. (**A**) Experimental design: embryos were collected at E2.5 and treated with ROCK inhibitor Y-27632 (ROCKi) or DMSO (control) for 24 hr. (**B–B'**) Confocal images of apical (PARD6B) and basolateral (CDH1) membrane components in control and ROCKi-treated embryos. As expected, PARD6B and CDH1 are mislocalized to the entire cell membrane of all cells in ROCKi-treated embryos, demonstrating effective ROCK inhibition (n = number of embryos examined). (**C–C'**) In control embryos, SOX2 is detected only in inside cells, while in ROCKi-treated embryos, SOX2 is detected in inside and outside cells (arrowheads, outside cells; n = embryos). (**D**) Quantification of ectopic SOX2 detected in outside cells of control and ROCKi-treated embryos (p, student's t-test, n = embryos). (**E**) SOX2 and CDX2 staining in outside cells of control and ROCKi-treated embryos. ROCK-inhibitor treatment leads to outside cells with mixed lineage marker expression (CDX2+/SOX2+). (**F**) Experimental design: embryos were collected at E1.5 and one of two blastomeres injected with mRNAs encoding YAP1^CA and GFP. Embryos were cultured for 72 hr, fixed, and then analyzed by immunofluorescence and confocal microscopy. (**G**) SOX2 is detected non-injected inside cells. SOX2 is not detected in YAP1^CA-overexpressing inside cells (arrowheads), n = embryos. (**H**) Across multiple embryos, all non-injected inside cells express SOX2, whereas the vast majority of YAP1^CA-injected inside cells fail to express SOX2.

DOI: https://doi.org/10.7554/eLife.42298.003

The following figure supplement is available for figure 1:

**Figure supplement 1.** Effect of ROCK1/2 inhibition on *Cdx2* expression and effect *Yap1^CA* overexpression on YAP1 localization and phosphorylation.
DOI: https://doi.org/10.7554/eLife.42298.004

conspicuous decrease in the proportion of $Yap1^{CA}$-overexpressing inside cells expressing detectable SOX2 (*Figure 1G,H*). Therefore, nuclear YAP1 is sufficient to repress *Sox2* expression in the inner cell mass, indicative of a likely role for YAP1 in repressing expression of *Sox2* in the trophectoderm downstream of ROCK1/2.

## LATS kinase is sufficient to induce inside cell positioning

To functionally test of the role of YAP1 in repressing expression of *Sox2*, we injected one of two blastomeres with mRNA encoding LATS2 kinase, which inactivates YAP1 and, presumably, the related protein WWTR1 by phosphorylation, causing their cytoplasmic retention (*Nishioka et al., 2008*). We then examined expression of SOX2 after culturing embryos to the blastocyst stage (*Figure 2A*), predicting that LATS2 kinase would induce the ectopic expression of *Sox2* in outside cells. Surprisingly, we observed that almost all *Lats2*-overexpressing cells ended up within the inner cell mass by the blastocyst stage (*Figure 2B,C*), in contrast to cells injected with *GFP* mRNA only, which contributed to both inner cell mass and trophectoderm. Notably, SOX2 was detected in all *Lats2*-overexpressing cells observed within the inner cell mass (*Figure 2D*), suggesting that *Lats2*-overexpressing cells were not only localized to the inner cell mass but also exhibited position-appropriate regulation of *Sox2*.

The strikingly increased prevalence of *Lats2*-overexpressing cells in the inner cell mass was also associated with a stark decrease in the number of *Lats2*-overexpressing cells detected within the trophectoderm and a decrease in the number of outside cells compared to embryos injected with *GFP* mRNA alone (*Figure 2C,E*), suggesting that *Lats2*-overexpressing outside cells either internalize or undergo cell death. Furthermore, we observed cellular fragments within the trophectoderm of *Lats2*-overexpressing embryos (*Figure 2B*, yellow arrowheads), as well as increased TUNEL staining in *Lats2*-overexpressing embryos compared to embryos injected with *GFP* mRNA only (*Figure 2—figure supplement 1A–B,D*), consistent with increased death of *Lats2*-overexpressing cells.

In addition to detecting SOX2 in all *Lats2*-overexpressing cells located inside the embryo, SOX2 was also detected in rare *Lats2*-overexpressing cells that remained on the embryo surface (*Figure 2D*). Therefore, LATS2 is sufficient to induce expression of SOX2 in cells regardless of their position within the embryo. We predicted that, if *Lats2* overexpression drove cells to adopt inner cell mass fate by influencing YAP1 and WWTR1 activity, then co-overexpression of $Yap1^{CA}$ would enable *Lats2*-overexpressing cells to contribute to trophectoderm. Consistent with this prediction, cooverexpression of *Lats2* and $Yap1^{CA}$ led to a significant decrease in the proportion of *Lats2*-overexpressing cells contributing to the inside cell position, and a concomitant increase in the proportion of *Lats2*-overexpressing cells remaining in the outside position (Figure *Figure 2—figure supplement 2A–D*). Moreover, cooverexpression of *Lats2* and $Yap1^{CA}$ reduced the number of TUNEL positive nuclei, consistent with $Yap1^{CA}$ rescuing survival of outside-positioned *Lats2*-overexpressing cells (*Figure 2—figure supplement 1C–D*). Collectively, these observations strongly suggest that LATS2 promotes inside cell positioning by regulating the activities of YAP1 and, likely, the related protein WWTR1.

To pinpoint when *Lats2*-overexpressing cells come to occupy the inside of the embryo, we performed a time course, examining the position of injected and non-injected cells from the 16-cell to the blastocyst stage (~80 cells). Surprisingly, between the 16 and 32-cell stages, the proportion of injected and non-injected cells in the total, outside, and inside cell populations were comparable whether embryos had been injected with *Lats2* and *GFP* or *GFP* mRNA alone (*Figure 2F–H*). In embryos injected with *GFP* mRNA alone, the proportion of injected and non-injected cells making up the total, outside, and inside cell populations remained constant throughout the time course. In contrast, starting around the 32-cell stage, the average proportion of *Lats2*-overexpressing cells making up the inside population began to increase dramatically. This increase was associated with a decrease in the proportion *Lats2*-overexpressing cells making up the outside population, consistent with internalization of *Lats2*-overexpressing cells after the 32-cell stage (*Figure 2G*). After the 32-cell stage, *Lats2*-injected cells became underrepresented as a proportion of the total cell population (*Figure 2H*), lending further support to the idea that *Lats2*-overexpressing cells that fail to internalize undergo cell death. Interestingly, the inside-skewed contribution of *Lats2*-overexpressing cells did not influence the ability of non-injected cells to contribute to the ICM (*Figure 2I*), arguing that *Lats2*-overexpression drives inside positioning cell-autonomously. We therefore conclude that *Lats2* overexpression acts on cell position and survival around the time of blastocyst formation.

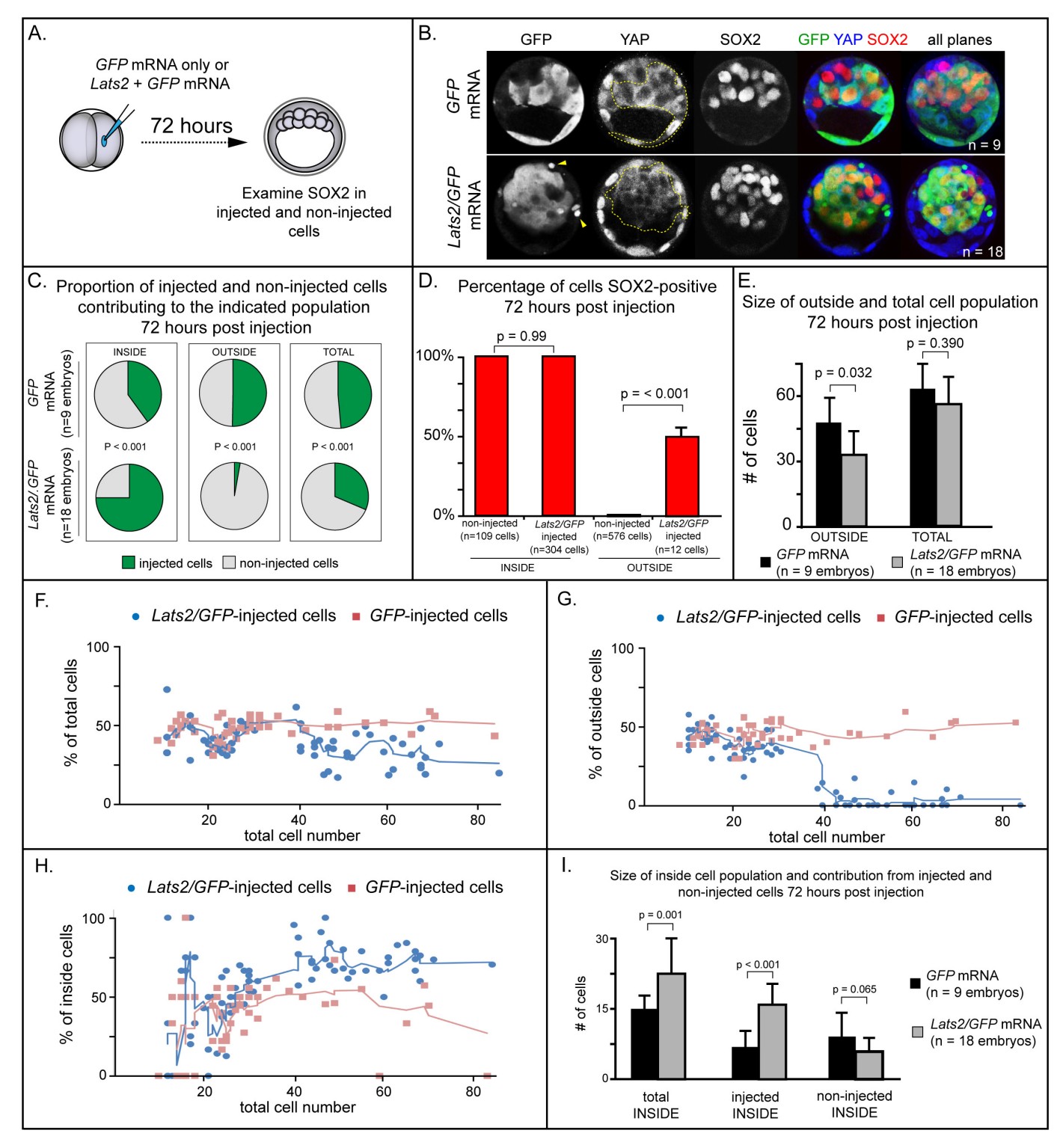

**Figure 2.** LATS2 kinase is sufficient to direct cells to inner cell mass fate. (A) Embryos were collected at E1.5 and one of two blastomeres was injected with mRNAs encoding LATS2 and GFP or GFP alone. Embryos were cultured for 72 hr, fixed, and then analyzed by immunofluorescence and confocal microscopy. (B) Cells injected with *GFP* (dotted line) contributed to trophectoderm and inner cell mass, while cells injected with *Lats2* and *GFP* (dotted line) contributed almost exclusively to the inner cell mass, leaving only cellular fragments in the trophectoderm (arrows), suggestive of cell death (n = embryos). (C) Proportion of inside, outside, and total cell populations across multiple embryos, which were comprised of non-injected cells, or cells injected with either *GFP* or *GFP/Lats2* mRNAs. Cells injected with *GFP/Lats2* were overrepresented within the inside cell population and

*Figure 2 continued on next page*

*Figure 2 continued*

underrepresented in the outside and total cell populations, relative to cells injected with *GFP* alone (P, chi-squared test). (**D**) Percentage of SOX2-positive cells within non-injected and *GFP*-injected or *Lats2/GFP*-injected populations observed inside and outside of the embryo. SOX2 was detected in all of the *Lats2/GFP*-injected inside cells, and in half of the rare, *Lats2/GFP*-injected outside cells (same number of embryos as in panel C) (p, student's t-test). (**E**) Average number of outside and total cells per embryo. The average number of outside cells is reduced in embryos injected with *Lats2/GFP*, relative to *GFP*-injected (p, student's t-test). (**F**) Proportion of *GFP* and *Lats2/GFP*-injected cells, relative to total cell number, over the course of development to the ~80-cell blastocyst (solid lines = average of indicated data point and four previous data points). (**G**) Data as shown in panel H, shown relative to outside cell number. (**H**) Data as shown in panel H, shown relative to inside cell number. (**I**) Contribution of injected and non-injected cells to the inside cell population, following injection with *GFP* or *Lats2/GFP*. Injection with *Lats2/GFP* increases the overall number of inside cells compared to injection with GFP only through increasing the number of injected cells contributing to the inside cell population, without affecting the number of non-injected cells contributing to the inside cell population (p, student's t-test).

DOI: https://doi.org/10.7554/eLife.42298.005

The following figure supplements are available for figure 2:

**Figure supplement 1.** *Lats2*-overexpressing cells die on the surface of the embryo (**A**) Merge of all confocal sections from TUNEL assay performed on an embryo injected with *GFP* mRNA into one blastomere at the two-cell stage and then cultured until the blastocyst stage.

DOI: https://doi.org/10.7554/eLife.42298.006

**Figure supplement 2.** LATS2 drives cells to an inside position by inhibiting YAP1 activity (**A–A'**) Cooverexpression of *Yap1*$^{CA}$ and *Lats2* partially rescues the ability of *Lats2*-overexpressing cells to contribute to trophectoderm and to repress *Sox2*.

DOI: https://doi.org/10.7554/eLife.42298.007

## LATS2 induces positional changes independent of *Sox2*

Our observation that *Lats2*-overexpression induces both the expression of SOX2 and cell repositioning to inner cell mass prompted us to investigate whether SOX2 itself drives cell repositioning downstream of *Lats2*. In support of this hypothesis, SOX2 activity has been proposed to bias inner cell mass fate (*Goolam et al., 2016*; *White et al., 2016*). We therefore investigated whether *Sox2* is required for the inner cell mass-inducing activity of LATS2 by overexpressing *Lats2* in embryos lacking maternal and zygotic *Sox2* (*Figure 3A*), as previously described (*Wicklow et al., 2014*). However, we observed that *Lats2*-overexpressing cells were equally likely to occupy inside position in the presence and absence of *Sox2* (*Figure 3B,C*). Furthermore, *Lats2*-overexpressing cells were equally unlikely to occupy outside position in the presence and absence of *Sox2* (*Figure 3D*). Therefore, although *Lats2* overexpression is sufficient to induce expression of *Sox2*, LATS2 acts on cell positioning/survival independently of *Sox2*.

## LATS2 antagonizes formation of the apical domain

Trophectoderm cell fate has been proposed to be determined by apically localized membrane components that maintain the position of future trophectoderm cells on the embryo surface (*Anani et al., 2014*; *Korotkevich et al., 2017*; *Maître et al., 2016*; *Maître et al., 2015*; *Samarage et al., 2015*; *Zenker et al., 2018*). For example, the apical membrane components aPKC and PARD6B are required for maintaining outside cell position and trophectoderm fate (*Alarcon, 2010*; *Dard et al., 2009*; *Hirate et al., 2015*; *Plusa et al., 2005*). Because *Lats2* overexpression led cells to adopt an inside position, this raised the testable possibility that LATS2 antagonizes localization of aPKC and PARD6B.

Since *Lats2* overexpression leads to cell positioning starting around the 32-cell stage, we examined the localization of aPKCz and PARD6B in embryos just prior to the 32-cell stage. At this stage, apical membrane components PARD6B and aPKCz were detected at the apical membrane of non-injected outside cells and outside cells injected with *GFP* only (*Figure 4A–D*). By contrast, most *Lats2*-overexpressing outside cells lacked detectable aPKCz and PARD6B (*Figure 4A–D*). Therefore, LATS2 is sufficient to antagonize localization of key apical domain proteins in outside cells, providing a compelling mechanism for the observed repositioning of *Lats2*-overexpressing outside cells.

We also examined other markers of apicobasal polarization in *Lats2*-overexpressing outside cells prior to the 32-cell stage. Curiously, other markers of apicobasal polarization were properly localized in all cells examined. For example, CDH1 was restricted to the basolateral membrane (*Figure 4E*), while filamentous Actin and phospho-ERM were restricted to the apical domain in outside cells of both *Lats2*-overexpressing and non-injected outside cells (*Figure 4F,G*). Thus, we propose that *Lats2*-overexpressing outside cells initially possess hallmarks of apicobasal polarization, but aPKC

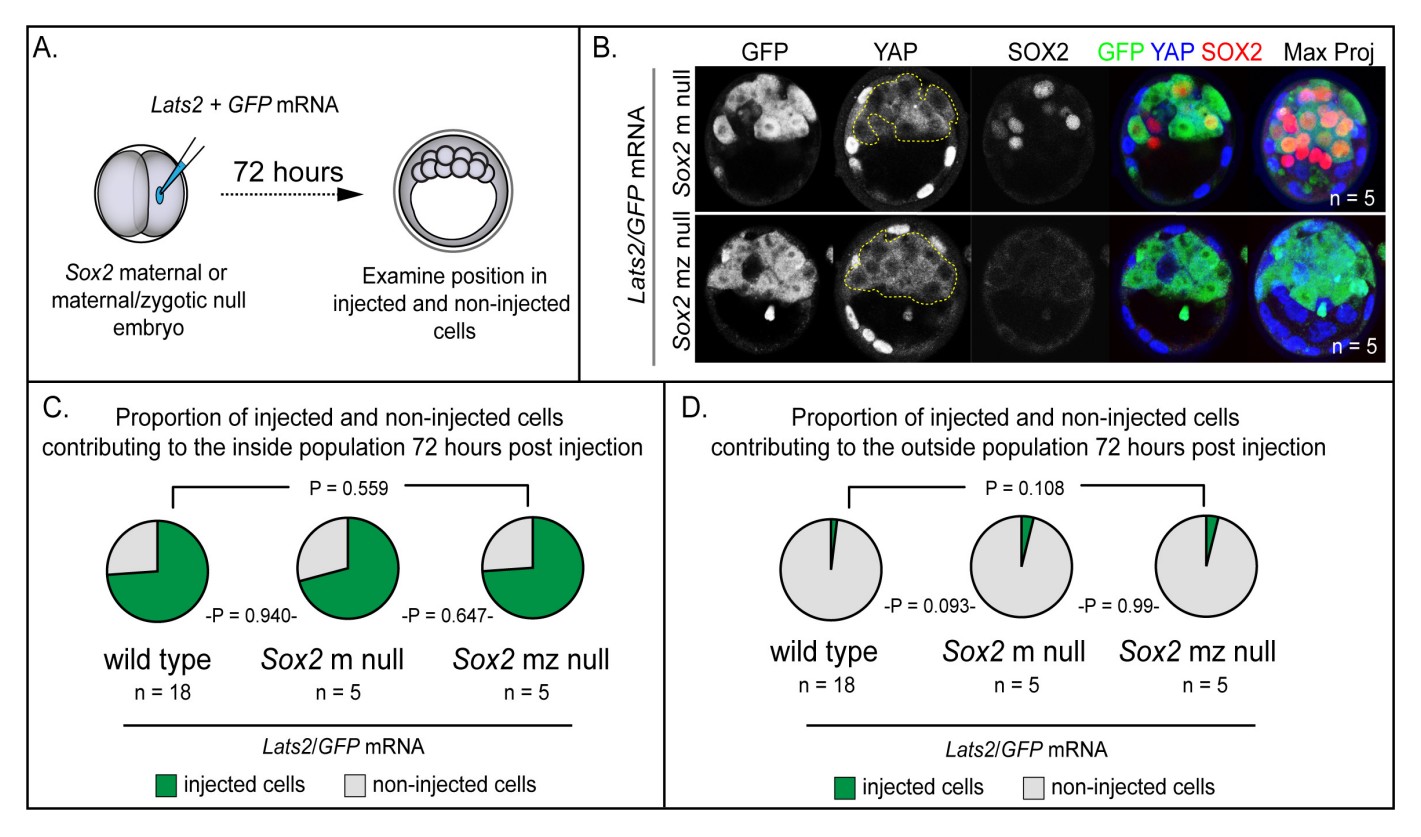

**Figure 3.** LATS2 directs inner cell mass fate independently of *Sox2* (A) *Lats2* and *GFP* or *GFP* alone were overexpressed in embryos lacking maternal or maternal and zygotic *Sox2*. (B) *Lats2/GFP*-overexpressing cells (dotted line) contribute almost exclusively to the inner cell mass in the presence or absence of *Sox2* (n = embryos). (C) Proportion of non-injected cells and cells injected with *Lats2/GFP* mRNAs contributing to inner cell mass in the indicated genetic backgrounds. No significant differences were observed based on embryo genotype, indicating that *Sox2* is dispensable for inside positioning by *Lats2*-overexpression (P, chi-squared test; n = embryos). (D) Proportion of non-injected cells and cells injected with the indicated mRNAs contributing to trophectoderm in the indicated genetic backgrounds. No significant differences were observed based on embryo genotype (P, chi-squared test; n = embryos).

DOI: https://doi.org/10.7554/eLife.42298.008

and PARD6B fail to properly localize, leading to the eventual depolarization and internalization of outside cells.

## YAP1 and WWTR1 restrict *Sox2* expression to the inner cell mass

Our overexpression data suggested that the activities of YAP1 and WWTR1 are important for regulating cell fate and gene expression. Next, we aimed to test the requirements for *Yap1* and *Wwtr1* in embryogenesis. *Yap1* null embryos survive until E9.0 (*Morin-Kensicki et al., 2006*), suggesting that oocyte-expressed (maternal) *Yap1* (*Yu et al., 2016*), or the *Yap1* paralogue *Wwtr1* (*Varelas et al., 2010*) are important for preimplantation development. However, embryos lacking maternal and zygotic *Wwtr1* and *Yap1* have not been scrutinized.

To generate embryos lacking maternal and zygotic *Wwtr1* and *Yap1*, we deleted *Wwtr1* and *Yap1* from the female germ line using mice carrying conditional alleles of *Wwtr1* and *Yap1* (*Xin et al., 2013*; *Xin et al., 2011*) and the female germ line-specific *Zp3Cre* (*de Vries et al., 2000*). We then crossed these females to males heterozygous for deleted alleles of *Wwtr1* and *Yap1* (see Materials and methods). From these crosses, we obtained embryos lacking maternally provided *Wwtr1* and *Yap1* and either heterozygous or null for *Wwtr1* and/or *Yap1* (*Supplementary file 1*). At E3.25 (≤32 cells), SOX2 and CDX2 are normally mutually exclusive (*Figure 5A*). However, with decreasing number of wild type zygotic alleles of *Wwtr1* and *Yap1*, we observed worsening phenotypes (*Figure 5B–F*). In the complete absence of *Wwtr1* and *Yap1*, we observed a severe loss of

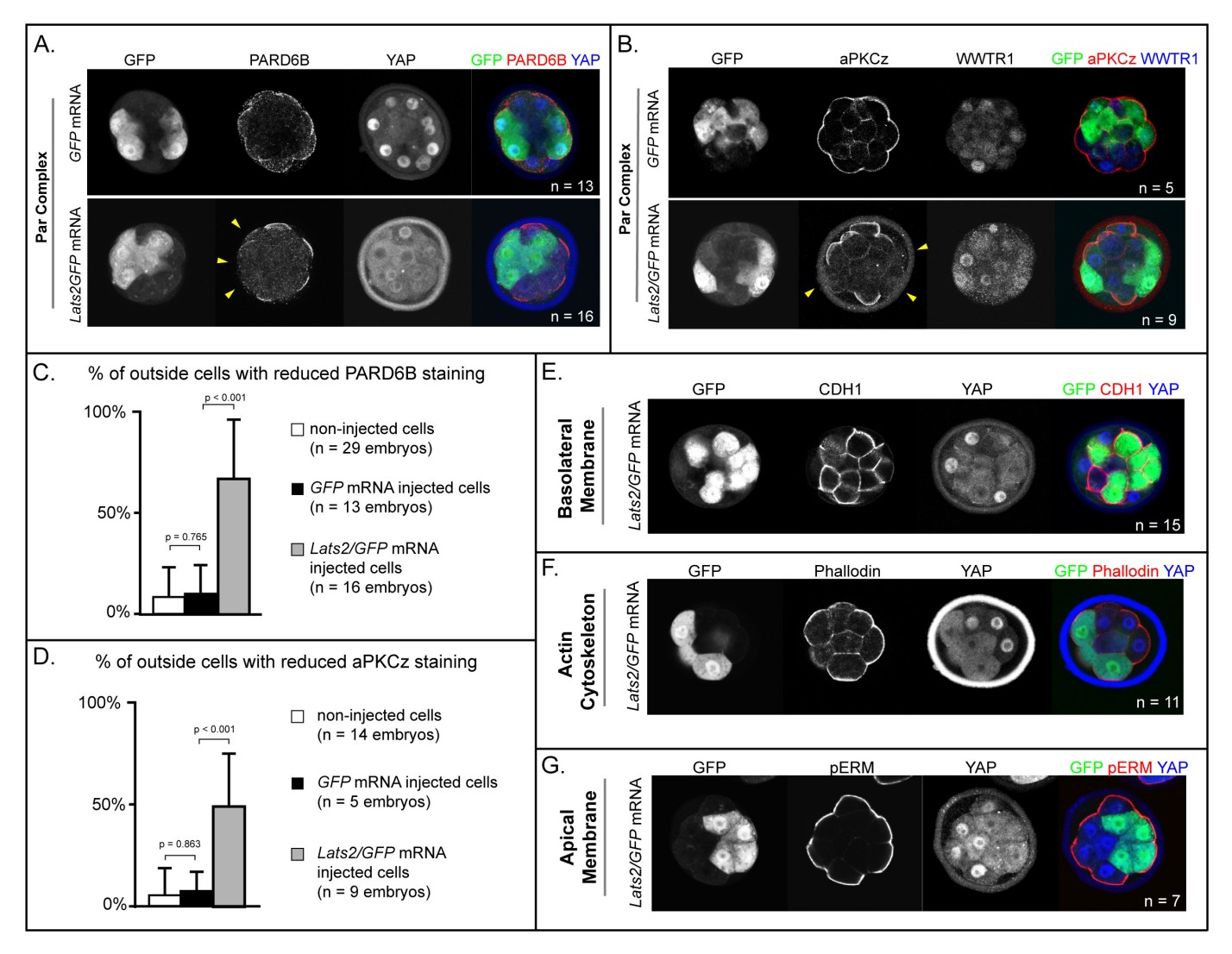

**Figure 4.** LATS2 antagonizes formation of the apical domain (A) In embryos at 16–32 cell stages, PARD6B is detectable in *GFP*-overexpressing and in non-injected cells, but not in *Lats2*-overexpressing cells (arrowheads, n = embryos). (B) At 16–32 cell stages, aPKCz is detectable in *GFP*-overexpressing and in non-injected cells, but not in *Lats2*-overexpressing cells (arrowheads, n = embryos). (C) Quantification of embryos shown in panel A (p, student's t-test). (D) Quantification of embryos shown in panel B (p, student's t-test). (E) At 16–32 cell stages, CDH1 is localized to the basolateral membrane in both *Lats2*-overexpressing and non-injected cells (n = embryos). (F) At 16–32 cell stages, Phalloidin staining demonstrates that filamentous Actin is apically enriched in *Lats2*-overexpressing and non-injected cells (n = embryos). (G) At 16–32 cell stages, pERM is localized to the apical membrane in both *Lats2*-overexpressing and non-injected cells (n = embryos).

DOI: https://doi.org/10.7554/eLife.42298.009

CDX2 and expansion of SOX2 in outside cells (*Figure 5D–F*), phenocopying *Lats2* overexpression. However, in embryos of intermediate genotypes, we observed expanded SOX2 and persistent, yet lower, expression levels of CDX2 (*Figure 5C,E–F*). Thus, regulation of *Sox2* expression is more sensitive to *Wwtr1* and *Yap1* dosage than is *Cdx2*. Moreover, these observations indicate that intermediate doses of *Wwtr1* and *Yap1* produce outside cells expressing markers of mixed cell lineage at E3.25.

## YAP1 and WWWTR1 maintain outside cell positioning

Based on our observations of *Lats2*-overexpressing embryos, we anticipated that defects in cell positioning in embryos lacking maternal and zygotic *Wwtr1* and *Yap1* could arise after the 32-cell stage.

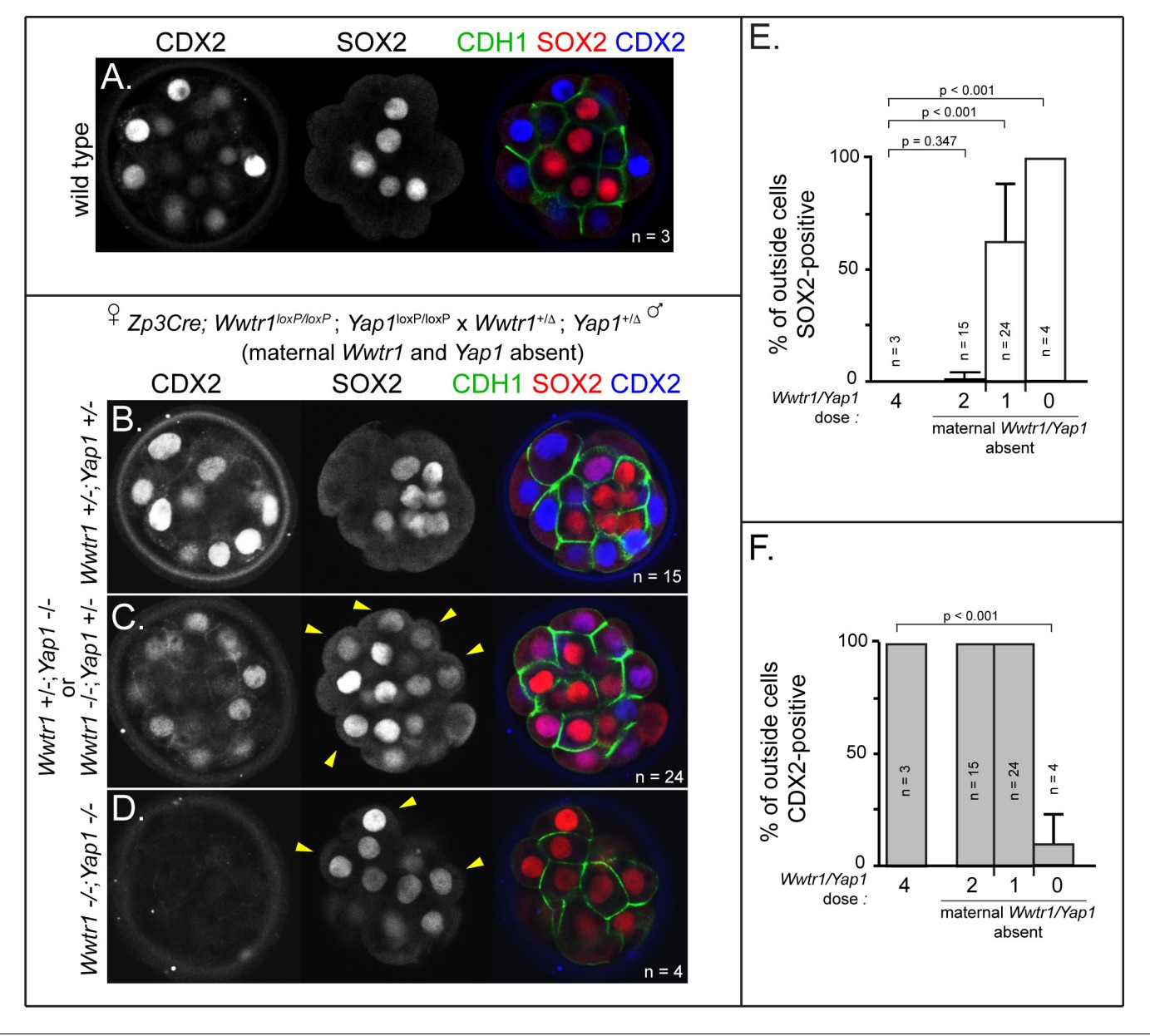

**Figure 5.** *Wwtr1* and *Yap1 are* required to repress SOX2 expression in outside cells. (A) CDX2 and SOX2 in wild type embryos at E3.25 (16–32 cell stages). CDX2 staining is more intense in outside cells than inside cells and SOX2 staining is specific to inside cells (n = embryos). (B) Embryos lacking maternal *Wwtr1* and *Yap1* with and heterozygous for *Wwtr1* and *Yap1* (which we consider to have 2 doses of WWTR1/YAP1) exhibit normal CDX2 and SOX2 expression (n = embryos). (C) Embryos lacking maternal *Wwtr1* and *Yap1* and heterozygous for either *Wwtr1* or *Yap1* (1 dose of WWTR1/YAP1) exhibit a high degree of ectopic SOX2 in outside cells (arrowheads), but continue to express CDX2, although the levels appear reduced (n = embryos). (D) Embryos lacking maternal and zygotic *Wwtr1* and *Yap1* (0 doses of WWTR1/YAP1) have the most severe phenotype, with a high degree of ectopic SOX2 in outside cells (arrowheads) and little or no detectable CDX2 (n = embryos). (E) Quantification of the percentage of outside cells in which ectopic SOX2 is detected in the presence of decreasing dose of *Wwtr1* and *Yap1* (t = student's t-test, n = embryos). (F) Quantification of the percentage of outside cells in which CDX2 is detected in the presence of decreasing dose of *Wwtr1* and *Yap1* (t = student's t-test, n = embryos).
DOI: https://doi.org/10.7554/eLife.42298.010

We therefore examined embryos lacking *Wwtr1* and *Yap1* at E3.75, when embryos possess more than 32 cells. Indeed, we observed skewed lineage contributions, correlating with the dosage of *Wwtr1* and *Yap1* (*Figure 6A–D*). Embryos with one or fewer wild type alleles of *Wwtr1* or *Yap1*

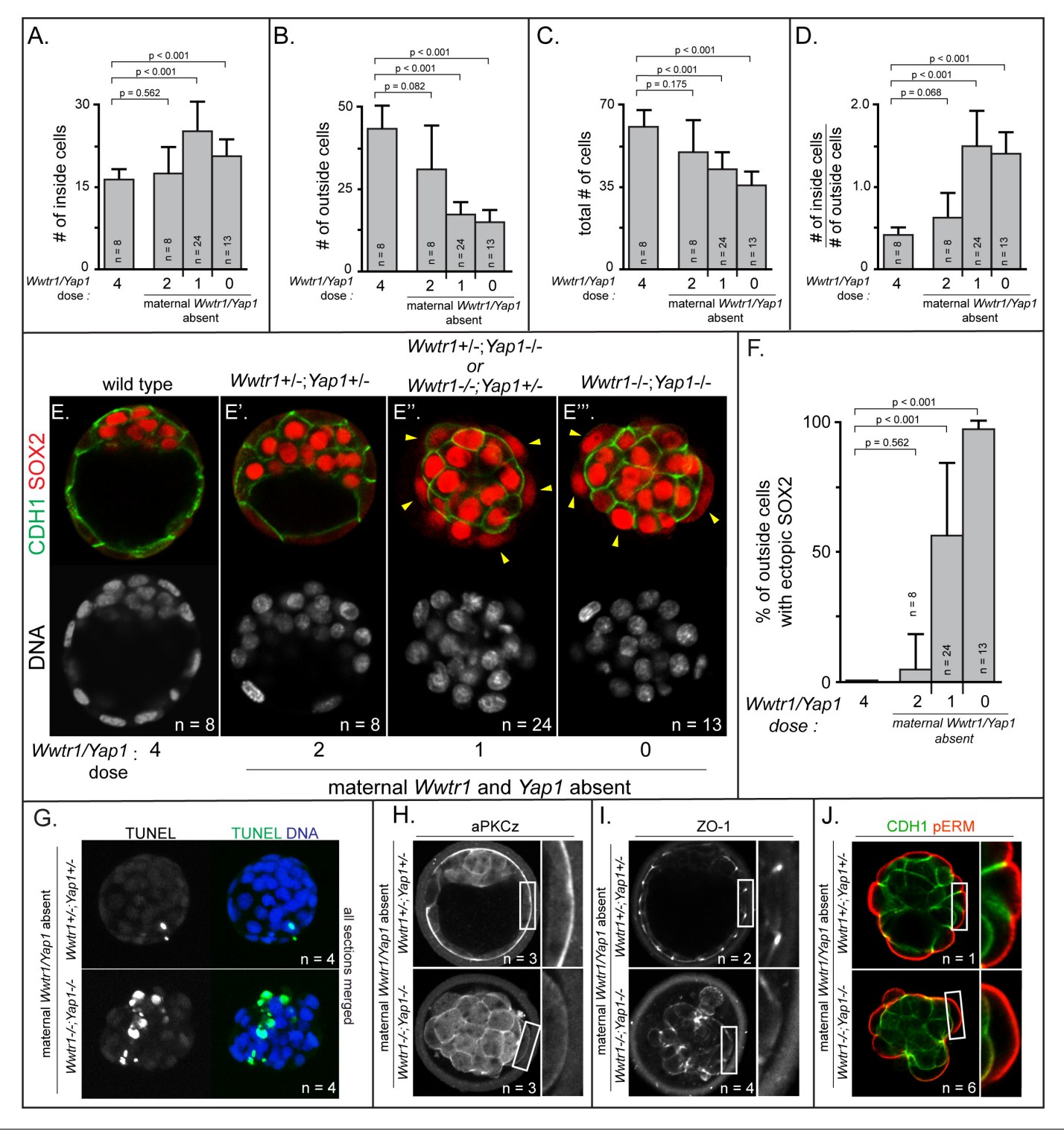

**Figure 6.** Positioning and epithelialization defects in embryos with *Wwtr1* and *Yap1* null alleles (**A**) Quantification of the average number of inside cells per embryo with decreasing dose of *Wwtr1* and *Yap1*. The number of inside cells increases as the dose of wild type *Wwtr1* and *Yap1* alleles is reduced (p, student's t-test, n = embryos). (**B**) Quantification of the average number of outside cells per embryo with decreasing dose of *Wwtr1 and Yap1*. The number of outside cells decreases as the dose of wild type *Wwtr1* and *Yap1* alleles is reduced (p, student's t-test, n = embryos). (**C**) Quantification of the average number of total cells per embryos with decreasing dose of wild type zygotic *Wwtr1* and *Yap1*. The number of total cells decreases as the dose of wild type *Wwtr1* and *Yap1* is reduced (p, student's t-test, n = embryos). (**D**) Quantification of the average ratio of inside to outside cells per embryo with decreasing dose of *Wwtr1* and *Yap1*. The ratio of inside to outside cells increases as the dose of wild type *Wwtr1* and *Yap1* is reduced (p,

*Figure 6 continued on next page*

*Figure 6 continued*

student's t-test, n = embryos). (E) Wild type embryos at E3.75 exhibit inner cell mass-specific expression of SOX2 (n = embryos). (E') E3.75 embryos lacking maternal *Wwtr1* and *Yap1* and heterozygous for zygotic *Wwtr1* and *Yap1* cavitate and repress *Sox2* in outside cells, leading to inner cell mass-specific expression of SOX2 similar to wild type embryos (n = embryos). (E'') Embryos lacking maternal *Wwtr1* and *Yap1* but with only one wild type allele of *Wwtr1* or *Yap1* fail to cavitate and repress *Sox2* in outside cells, leading to ectopic SOX2 in outside cells (arrowheads, n = embryos). (E''') Embryos lacking maternal and zygotic *Wwtr1* and *Yap1* fail to cavitate and repress *Sox2* in outside cells, leading to ectopic SOX2 in outside cells (arrowheads, n = embryos). (F) Quantification of ectopic SOX2 detected in embryos such as those shown in panels E-E''. The percentage of outside cells with ectopic SOX2 increases as the dose of wild type *Wwtr1* and *Yap1* alleles is reduced (p, student's t-test, n = embryos). (G) TUNEL analysis of embryos lacking maternal *Wwtr1* and *Yap1* heterozygous for zygotic *Wwtr1* and *Yap1* or lacking maternal and zygotic *Wwtr1* and *Yap1*. Extensive TUNEL staining is observed in embryos lacking maternal and zygotic *Wwtr1* and *Yap1* indicative of cell death. Max projections of all confocal sections from a single embryo are shown (n = embryos). (H) aPKCz staining in embryos lacking maternal *Wwtr1* and *Yap1*, either heterozygous for zygotic *Wwtr1* and *Yap1* or with no zygotic *Wwtr1* and *Yap1*. aPKC is not localized to the apical membrane of embryos with no zygotic *Wwtr1* and *Yap1* (n = embryos). (I) ZO-1 staining in embryos lacking maternal *Wwtr1* and *Yap1*, either heterozygous for zygotic *Wwtr1* and *Yap1* or with no zygotic *Wwtr1* and *Yap1*. ZO-1 is disorganized in embryos with no zygotic *Wwtr1* and *Yap1*, suggesting that formation of a mature epithelium depends on *Wwtr1* and *Yap1* (n = embryos). (J) pERM and CDH1 staining in embryos lacking maternal *Wwtr1* and *Yap1*, either heterozygous for zygotic *Wwtr1* and *Yap1* or with no zygotic *Wwtr1* and *Yap1*. pERM is localized to apical membranes and CDH1 to basolateral membranes regardless of the dose of wild type *Wwtr1* and *Yap1* alleles (n = embryos).

DOI: https://doi.org/10.7554/eLife.42298.011

The following figure supplement is available for figure 6:

**Figure supplement 1.** Increased cell death and epithelialization defects in embryos lacking maternal *Wwtr1* and *Yap1* with a single wild type allele of *Wwtr1* or *Yap1*.

DOI: https://doi.org/10.7554/eLife.42298.012

exhibited an increase in the number of inside cells and a reduction in the number of outside cells (*Figure 6A–B*), consistent with altered cell positioning.

Although the average total number of cells was also reduced in these embryos (*Figure 6C*), the reduction in total cell number did not alone account for the loss of cells on the outside of the embryo (*Supplementary file 2*). This observation suggested that, similar to *Lats2*-overexpressing cells, cells with reduced *Wwtr1* and *Yap1* exhibit an increased frequency of outside cell death, in addition to increased outside cell internalization. Consistent with this, embryos with one or fewer wild type alleles of *Wwtr1* or *Yap1* exhibited an increase in the ratio of inside to outside cells (*Figure 6D*) and an increase in cells undergoing apoptosis by TUNEL assay (*Figure 6G* and *Figure 6—figure supplement 1A,B*).

Critically, the fewer outside cells in embryos lacking *Wwtr1* and *Yap1*, which appeared stretched over the mass of inside cells, exhibited ectopic expression of SOX2 (*Figure 6E–F*). Therefore, WWTR1/YAP1 repress inner cell mass fate, downstream of LATS kinases. Intriguingly, our data also indicate that WWTR1 is a more potent repressor of *Sox2* at E3.75 than YAP1 since embryos with a single wild type allele of *Wwtr1* had significantly fewer cells expressing ectopic SOX2 then embryos with a single wild type allele of *Yap1* (*Figure 6—figure supplement 1F,G*).

Since loss of *Wwtr1* and *Yap1* phenocopied *Lats2* overexpression in terms of *Sox2* expression, cell death, and cell repositioning, we next evaluated the apical domain and cell polarization in outside cells of embryos lacking *Wwtr1* and *Yap1* at E3.75. As expected, observed greatly reduced aPKC at the apical membrane of outside cells in embryos with one or fewer doses of *Wwtr1* or *Yap1* (*Figure 6H* and *Figure 6—figure supplement 1C*). In addition, we evaluated the localization of the tight junction protein ZO-1, which suggested failure in tight junction formation in embryos with one or fewer doses of *Wwtr1* and *Yap1* (*Figure 6I* and *Figure 6—figure supplement 1D*). Notably, however, other markers of apicobasal polarity, such as CDH1 and pERM were correctly localized in outside cells of mutant embryos at this stage (*Figure 6J* and *Figure 6—figure supplement 1E*), consistent with some normal cell polarization. Our observations indicate that WWTR1 and YAP1 play a crucial role in the formation of the apical domain and maintaining the positioning and survival of outside cells while repressing expression of *Sox2*.

## Discussion

During preimplantation development, lineage-specific transcription factors are commonly expressed in 'noisy' domains before refining to a lineage-appropriate pattern (*Simon et al., 2018*). For

example, *Oct4* and *Nanog* are expressed in both inner cell mass and trophectoderm until after blastocyst formation (*Dietrich and Hiiragi, 2007*; *Strumpf et al., 2005*). Similarly, CDX2 is detected in inner cell mass, as well as trophectoderm, until blastocyst stages (*McDole and Zheng, 2012*; *Ralston and Rossant, 2008*; *Strumpf et al., 2005*). In striking contrast to these genes, SOX2 is never detected in outside cells (*Wicklow et al., 2014*), indicating that robust mechanisms must exist to minimize noise and prevent its aberrant expression in trophectoderm. Here, we identify YAP1/WWTR1 as key components that repress *Sox2* expression in outside cells of the embryo. Notably, manipulations known to antagonize YAP1/WWTR1 activity, including chemical inhibition of ROCK and overexpression of LATS2, lead to ectopic expression of SOX2 in outside cells, reinforcing the notion that YAP1/WWTR1 activity are crucial for repression of *Sox2* in outside cells.

Additionally, we find that *Sox2* expression is more sensitive than is *Cdx2* to YAP1/WWTR1 activity, since intermediate doses of active YAP1/WWTR1 yield cells that coexpress both SOX2 and CDX2 (*Figure 7A*). This observation is consistent with the fact that CDX2 is initially detected in inside cells of the embryo during blastocyst formation (*Dietrich and Hiiragi, 2007*; *McDole and Zheng, 2012*; *Ralston and Rossant, 2008*), where SOX2 is also expressed (*Wicklow et al., 2014*). Thus, inside cells could initially possess intermediate doses of active YAP1/WWTR1 at this early stage. By contrast, outside cells have greatly reduced YAP1/WWTR1 activity, owing to elevated LATS activity. In this way, the HIPPO pathway ensures robust developmental transitions, by rapidly nudging SOX2-expressing cells into their correct and final positions inside the embryo (*Figure 7B*).

Consistent with our proposed model, the timing of HIPPO-induced cell internalization coincides with loss of cell fate plasticity around the 32-cell stage (*Posfai et al., 2017*). This timing also coincides with the formation of mature tight junctions among outside cells (*Sheth et al., 1997*), which reinforce and intensify differences in HIPPO signaling activity between inside and outside compartments of the embryo (*Hirate and Sasaki, 2014*; *Leung and Zernicka-Goetz, 2013*). Our observations indicate that HIPPO signaling can, in turn, interfere with trophectoderm epithelialization. Therefore, we propose that HIPPO engages in a negative feedback loop with cell polarity components (*Figure 7B*).

We propose two mechanisms by which HIPPO signaling eliminates cells from the trophectoderm, both of which are downstream of YAP1/WWTR1 (*Figure 7C*). First, a small proportion of conflicted cells undergo cell death. This is in line with the observed increase in the level of apoptosis detected after the 32-cell stage (*Copp, 1978*). We showed that cell lethality due to elevated HIPPO can be rescued by increasing levels of nuclear YAP1, suggesting that YAP1 activity normally provides a pro-survival signal to trophectoderm cells, consistent with the proposed role of YAP1 in promoting proliferation in non-eutherian mammals (*Frankenberg, 2018*). Moreover, deletion of *Sox2* did not rescue survival of outside cells in which HIPPO signaling was artificially elevated, arguing that HIPPO resolves cell fate conflicts independently of lineage-specific genes.

The second way that conflicted cells are eliminated from the trophectoderm is that cells with elevated HIPPO signaling drive their own internalization. This is consistent with the observation that cells in which *Tead4* has been knocked down become internalized (*Mihajlović et al., 2015*). However, in contrast to *Tead4* loss of function, which preserves the apical domain in outside cells (*Mihajlović et al., 2015*; *Nishioka et al., 2008*), we observed that *Yap1/Wwtr1* loss of function leads loss of apical PARD6D/aPKC. These observations suggest that YAP1/WWTR1 could partner with proteins other than TEAD4 to regulate apical domain formation. Consistent with this proposal, TEAD1 has been proposed to play an essential role in the early embryo (*Sasaki, 2017*). Nevertheless, since PARD6B/aPKC are essential for outside cell positioning (*Dard et al., 2009*; *Hirate et al., 2015*; *Plusa et al., 2005*), the loss of the apical domain could affect cell positioning in several ways. For instance, loss of PARD6B/aPKC would eventually lead to cell depolarization (*Alarcon, 2010*), which could influence any of the processes normally governing the allocation of inside cells, such as oriented cleavage, cell contractility, or apical constriction (*Korotkevich et al., 2017*; *Maître et al., 2016*; *Samarage et al., 2015*). Identifying the downstream mechanisms by which HIPPO drives cells to inner cell mass will be a stimulating topic of future study.

Our studies also revealed that SOX2 does not play a role in cell positioning. This observation sheds light on a recent study, which showed that SOX2 dwells longer in select nuclei of four-cell stage embryos that are destined to contribute to the inner cell mass (*White et al., 2016*). We propose that SOX2 is associated with future pluripotent state but does not alone contribute to all aspects of pluripotency, such as inside positioning. It is therefore still unclear why it is important to

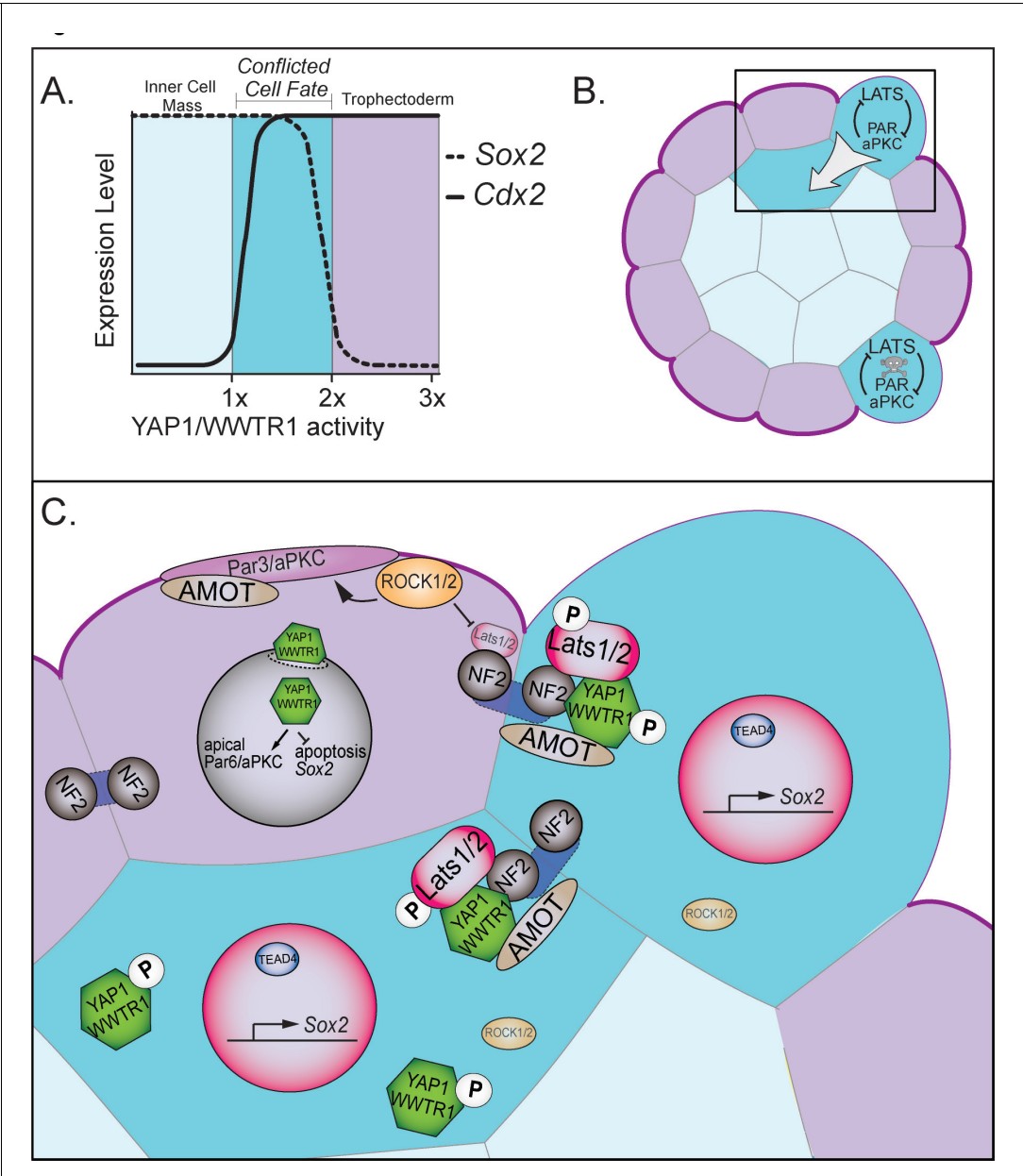

**Figure 7.** Resolution of cell fate conflicts in the preimplantation mouse embryo. (**A**) The expression of *Sox2* and *Cdx2* is differentially sensitive to YAP1/WWTR1 activity, leading to co-expression of both lineage markers in cells when YAP1/WWTR1 activity levels are intermediate. (**B**) During division from the 16 to the 32-cell stage, cells that inherit the apical membrane repress HIPPO signaling and maintain an outside position. However, cells that inherit a smaller portion of the apical membrane would initially elevate their HIPPO signaling. We propose that elevated HIPPO then feeds back onto polarity by further antagonizing PAR-aPKC complex formation, leading to a snowball effect on repression of *Sox2* expression, and thus ensuring that SOX2 is never detected in outside cells because these cells are rapidly internalized or apoptosed. (**C**) A closeup of the boxed region in panel B. In most outside cells, low LATS2 activity enables high levels of YAP1/WWTR1 activity, which repress *Sox2* and apoptosis and promote *Cdx2* expression and apical localization of aPKC and PARD6B, which in turn repress the HIPPO pathway. In rare outside cells, LATS2 activity becomes elevated, leading to lower activity of YAP1/WWTR1, which then leads these cells to become internalized or to undergo apoptosis.

DOI: https://doi.org/10.7554/eLife.42298.013

establish the inside cell-specific SOX2 expression during embryogenesis. Identification pathways that function downstream of YAP1/WWTR1 and in parallel to SOX2 to promote formation of pluripotent cells will provide meaningful insights into the natural origins of mammalian pluripotent stem cell progenitors.

# Materials and methods

**Key resources table**

| Reagent type (species) or resource | Designation | Source or reference | Identifiers | Additional information |
|---|---|---|---|---|
| Strain, strain background (*Mus musculus*) | CD-1 | Charles River Laboratories | RRID:IMSR _CRL:22 | |
| Strain, strain background (*M. musculus*) | $Sox2^{tm1.1Lan}$ | *Smith et al. (2009)* PMID:19666824 | RRID:IMSR_ JAX:013093 | mixed background, *Sox2* null refers to recombined allele |
| Strain, strain background (*M. musculus*) | *Wwtr1* conditional allele; $Wwtr1^{tm1.1Eno}$; $Wwtr1^{loxp}$ | *Xin et al., 2013* *Xin et al., 2013* PMID:23918388 | MGI:5544289 | mixed background, 'Wwtr1-' or '$Wwtr1^{\Delta}$' refers to recombined allele |
| Strain, strain background (*M. musculus*) | *Yap1* conditional allele; $Yap1^{tm1.1Eno}$; $Yap1^{loxp}$ | *Xin et al., 2011* *Xin et al., 2011* PMID:22028467 | MGI:5446483 | mixed background, , 'Yap1-' or '$Yap1^{\Delta}$' refers to recombined allele |
| Strain, strain background (*M. musculus*) | Tg(Zp3-cre) 93Knw; Zp3Cre | *de Vries et al., 2000* *de Vries et al., 2000* *de Vries et al., 2000* PMID:10686600 | RRID:MGI :3835503 | mixed background |
| Strain, strain background (*M. musculus*) | 129-Alpl $^{tm(cre)Nagy}$ | *Lomelí et al., 2000* *Lomelí et al., 2000* *Lomelí et al., 2000* PMID:10686602 | RRID:IMSR_ JAX:008569 | mixed background |
| Antibody | mouse anti-CDX2 | Biogenex | BioGenex Cat# AM392; RRID:AB_2650531 | (1:200) |
| Antibody | goat anti -SOX2 | Neuromics | Neuromics Cat# GT15098; RRID:AB_2195800 | (1:200) |
| Antibody | rabbit anti-PARD6B | Novus Biologicals | Novus Cat# NBP1-87337; RRID:AB_11034389 | (1:100) |
| Antibody | rabbit anti-PARD6B | Santa Cruz Biotechnology | Santa Cruz Biotechnology Cat# sc-67393; RRID:AB_2267889 | (1:100) |
| Antibody | mouse anti-PKCζ | Santa Cruz Biotechnology | Santa Cruz Biotechnology Cat# sc-17781; RRID:AB_628148 | (1:100) |
| Antibody | mouse anti-YAP1 | Santa Cruz Biotechnology | Santa Cruz Biotechnology Cat# sc-101199; RRID:AB_1131430 | (1:200) |
| Antibody | mouse anti-pYAP1 | Cell Signaling Technology | Cell Signaling Technology Cat# 4911; RRID:AB_2218913 | (1:800) |
| Antibody | chicken anti-GFP | Aves Labs | Aves Labs Cat# GFP-1020; RRID:AB_10000240 | (1:2000) |

*Continued on next page*

*Continued*

| Reagent type (species) or resource | Designation | Source or reference | Identifiers | Additional information |
|---|---|---|---|---|
| Antibody | rat anti-CDH1 | Sigma-Aldrich | Sigma-Aldrich Cat# U3254; RRID:AB_477600 | (1:500) |
| Antibody | mouse anti-ZO1 | Thermo Fisher Scienctific | Thermo Fisher Scientific Cat# 33–9100; RRID:AB_2533147 | (1:1000) |
| Recombinant DNA reagent | *Lats2* mRNA; LATS2 | *Nishioka et al., 2009* *Nishioka et al., 2009* *Nishioka et al., 2009* PMID:19289085 | pcDNA3.1 -pA83-Lats2; RIKEN: RDB12200 | In Vitro Transcription template for *Lats2* mRNA |
| Recombinant DNA reagent | *Yap1*$^{CA}$ mRNA; YAP1$^{CA}$ | *Nishioka et al., 2009* *Nishioka et al., 2009* *Nishioka et al., 2009* PMID:19289085 | pcDNA3.1-pA83 -HA-Yap-S112A; RIKEN: RDB12194 | In Vitro Transcription template for *Yap1CA* mRNA |
| Recombinant DNA reagent | *nls-GFP* mRNA; nls-GFP | *Ariotti et al., 2015* *Ariotti et al., 2015* *Ariotti et al., 2015* PMID:26585296 | Addgene: Plasmid #67652 | In Vitro Transcription template for nls-*GFP* mRNA |
| Recombinant DNA reagent | pCS2-EGFP; *EGFP mRNA*; GFP mRNA; GFP | *Chazaud et al., 2006* PMID: 16678776 | | In Vitro Transcription template for GFP mRNA |
| Commercial assay or kit | mMessage mMachine Sp6 Transcription Kit | Thermo Fisher Scienctific | Thermo Fisher Scientific Cat# AM1340 | |
| Commercial assay or kit | mMessage mMachine T7 Transcription Kit | Thermo Fisher Scienctific | Thermo Fisher Scientific Cat# AM1344 | |
| Commercial assay or kit | MEGAClear Transcription Clean-Up Kit | Thermo Fisher Scienctific | Thermo Fisher Scientific Cat# AM1908 | |
| Commercial assay or kit | In-Situ Cell Death Detection Kit, Fluorescein; TUNEL assay | Sigma-Aldrich | Sigma-Aldrich Cat# 11684795910 | |
| Commercial assay or kit | Extract-N -Amp Kit | Sigma-Aldrich | Sigma-Aldrich Cat # XNAT2 | |
| Chemical compound, drug | Y-27632; ROCK-inhibitor | Millipore | Millipore Cat# SCM075 | |
| Software, algorithm | Adobe Photoshop | Adobe | RRID:SCR_014199 | |
| Software, algorithm | Fiji | http://fiji.sc | RRID:SCR_002285 | |

## Mouse strains and genotyping

All animal research was conducted in accordance with the guidelines of the Michigan State University Institutional Animal Care and Use Committee. Wild type embryos were derived from CD-1 mice (Charles River). The following alleles or transgenes were used in this study, and maintained in a CD-1 background: *Sox2*$^{tm1.1Lan}$ (*Smith et al., 2009*), *Yap*$^{tm1.1Eno}$ (*Xin et al., 2011*), *Wwtr1*$^{tm1.1Eno}$ (*Xin et al., 2013*), *Tg(Zp3-cre)93Knw* (*de Vries et al., 2000*). Null alleles were generated by breeding mice carrying floxed alleles and mice carrying ubiquitously expressed *Cre*, *129-Alpl*$^{tm(cre)Nagy}$ (*Lomelí et al., 2000*).

## Embryo collection and culture

Mice were maintained on a 12 hr light/dark cycle. Embryos were collected by flushing the oviduct or uterus with M2 medium (Millipore). For embryo culture, KSOM medium (Millipore) was equilibrated overnight prior to embryo collection. Y-27632 (Millipore) was included in embryo culture medium at a concentration of 80 μM with 0.4% DMSO, or 0.4% DMSO as control, where indicated. Embryos were cultured at 37°C in a 5% $CO_2$ incubator under light mineral oil.

## Embryo microinjection

*Lats2* and *Yap1^{S112A}* (*Yap1^{CA}*) mRNA was synthesized from cDNAs cloned into the pcDNA3.1-poly (A)83 plasmid (*Yamagata et al., 2005*) using the mMESSAGE mMACHINE T7 transcription kit (Invitrogen). *EGFP* or *nls-GFP* mRNA were synthesized from *EGFP* cloned into the pCS2 plasmid or the *nls-GFP* plasmid (*Ariotti et al., 2015*) using the mMESSAGE mMACHINE SP6 transcription kit (Invitrogen). mRNAs were cleaned and concentrated prior to injection using the MEGAclear Transcription Clean-Up Kit (Invitrogen). *Lats2* and *YAP1^{CA}* mRNAs were injected into one blastomere of two-cell stage embryos at a concentration of 500 ng/μl, mixed with 350 ng/μl *EGFP* or *nls-GFP* mRNA diluted in 10 mM Tris-HCl (pH 7.4), 0.1 mM EDTA.

## Immunofluorescence and confocal microscopy

Embryos were fixed with 4% formaldehyde (Polysciences) for 10 min, permeabilized with 0.5% Triton X-100 (Sigma Aldrich) for 30 min, and then blocked with blocking solution (10% Fetal Bovine Serum (Hyclone), 0.1% Triton X-100) for 1 hr at room temperature, or overnight at 4°C. Primary Antibodies used were: mouse anti-CDX2 (Biogenex, CDX2-88), goat anti-SOX2 (Neuromics, GT15098), rabbit anti-PARD6B (Santa Cruz, sc-67393), rabbit anti-PARD6B (Novus Biologicals, NBP1-87337), mouse anti-PKCζ (Santa Cruz Biotechnology, sc-17781), rat anti-CDH1 (Sigma Aldrich, U3254), mouse anti-ZO1 (Thermo Fisher Scientific, 33–9100), mouse anti-YAP (Santa Cruz Biotechnology, sc101199), rabbit anti phospho-YAP (Cell Signaling Technologies, 4911), chicken anti-GFP (Aves, GFP-1020). Stains used were: Phallodin-633 (Invitrogen), DRAQ5 (Cell Signaling Technologies) and DAPI (Sigma Aldrich). Secondary antibodies conjugated to DyLight 488, Cy3 or Alexa Flour 647 fluorophores were obtained from Jackson ImmunoResearch. Embryos were imaged using an Olympus FluoView FV1000 Confocal Laser Scanning Microscope system with 20x UPlanFLN objective (0.5 NA) and 5x digital zoom. For each embryo, z-stacks were collected, with 5 μm intervals between optical sections. All embryos were imaged prior to knowledge of their genotypes.

## Embryo analysis

For each embryo, z-stacks were analyzed using Photoshop or Fiji, which enabled the virtual labeling, based on DNA stain, of all individual cell nuclei. Using this label to identify individual cells, each cell in each embryo was then assigned to relevant phenotypic categories, without knowledge of embryo genotype. Phenotypic categories included marker expression (e.g., SOX2 or CDX2 positive or negative), protein localization (e.g., aPKC or CDH1 apical, basal, absent, or unlocalized), and cell position, where cells making contact with the external environment were considered 'outside' and cells surrounded by other cells were considered 'inside' cells.

## TUNEL assay

Embryos were fixed, permeabilized, and blocked as described for immunofluorescence. Zonae pellucida were removed using Tyrode's Acid treatment prior to performing the TUNEL assay (In Situ Cell Death Detection Kit, Fluorescein, Millipore-Sigma). Embryos were incubated in 200 μl of a 1:10 dilution of enzyme in label solution for 2 hr at 37°C. Embryos were then washed twice with blocking solution for 10 min each, and then mounted in a 1 to 400 dilution of DRAQ5 in blocking solution to stain DNA.

## Embryo genotyping

To determine embryo genotypes, embryos were collected after imaging and genomic DNA extracted using the Extract-N-Amp kit (Sigma) in a final volume of 10 μl. Genomic extracts (1–2 μl) were then subjected to PCR using allele-specific primers (*Supplementary file 3*).

## Acknowledgements

We are grateful to Dr. Hiroshi Sasaki for providing expression constructs, to Dr. Randy L Johnson for providing mice carrying conditional alleles of *Yap1* and *Wwtr1*, and to Dr. Jason Knott for embryo microinjection training. We also thank Dr. Ripla Arora, Dr. Julia Ganz, and members of the Ralston Lab for comments. This work was supported by NIH R01 GM104009 and the Eunice Kennedy Shriver National Institute of Child Health and Human Development of the National Institutes of Health under Award Number T32HD087166. The content is solely the responsibility of the authors and does not necessarily represent the official views of the National Institutes of Health. We thank anonymous reviewers for insightful questions and suggestions.

## Additional information

### Funding

| Funder | Grant reference number | Author |
| --- | --- | --- |
| National Institute of General Medical Sciences | R01 GM104009 | Amy Ralston |
| National Institute of Child Health and Human Development | T32HD087166 | Tayler M Murphy |

The funders had no role in study design, data collection and interpretation, or the decision to submit the work for publication.

### Author contributions

Tristan Frum, Data curation, Formal analysis, Investigation; Tayler M Murphy, Data curation, Formal analysis; Amy Ralston, Conceptualization, Formal analysis, Funding acquisition, Investigation, Project administration

### Author ORCIDs

Tristan Frum http://orcid.org/0000-0003-3452-8762
Amy Ralston http://orcid.org/0000-0003-3755-8262

### Decision letter and Author response

Decision letter https://doi.org/10.7554/eLife.42298.019
Author response https://doi.org/10.7554/eLife.42298.020

## Additional files

### Supplementary files

• Supplementary file 1. Summary of embryos recovered from *Wwtr1;Yap1* germline null females.
DOI: https://doi.org/10.7554/eLife.42298.014

• Supplementary file 2. Mean and standard deviation of cell counts for every experimental treatment.
DOI: https://doi.org/10.7554/eLife.42298.015

• Supplementary file 3. Allele-specific primers used for determining embryo and mouse genotypes.
DOI: https://doi.org/10.7554/eLife.42298.016

• Transparent reporting form
DOI: https://doi.org/10.7554/eLife.42298.017

### Data availability

All data appear within the manuscript and associated files.

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
