## [Decision Letter]

[Editors’ note: a previous version of this study was rejected after peer review, but the authors submitted for reconsideration. The first decision letter after peer review is shown below.]

Thank you for submitting your work entitled "HIPPO provides a fail-safe for resolving embryonic cell fate conflicts during establishment of pluripotency in vivo" for consideration by *eLife*. Your article has been reviewed by two peer reviewers, and the evaluation has been overseen by a Reviewing Editor and a Senior Editor. The reviewers have opted to remain anonymous.

Our decision has been reached after consultation between the reviewers. Based on these discussions and the individual reviews below, we regret to inform you that in its current state your manuscript will not be considered further for publication in *eLife*.

However as you will see both reviewers were very positive and feel that your genetic approach certainly has provided further insights into lineage segregation in the early embryo. However the biggest issue is that the study is a bit premature given the small numbers of embryos you've been able to analyse to date. If you can considerably improve the manuscript along the lines suggested by the reviewers, including expanding the samples sizes for critical genotypes we would be happy to see a major revision of the paper in the coming months, which would be considered as a new submission.

*Reviewer #2:*

In this paper, the authors investigate further the mechanism of restriction of *Sox2* expression to the ICM during early mouse development. As the blastocyst forms, there is a segregation of expression of key lineage-specific transcription factors for the outer trophectoderm (TE) (Cdx2) and the pluripotent ICM (*Sox2*), which is regulated by the Hippo signaling pathway. They previously showed that *Sox2* is activated (probably indirectly) and Cdx2 inactivated by active Lats kinase function in the inside cells that will form the ICM. *Sox2* is regulated independently of Cdx2. Here they explore further this regulatory interaction by making floxed alleles of both *Yap* and *Wwtr1*, the redundant coactivators downstream of Hippo signaling. This enabled them to make a dosage series of maternal and zygotically inactivated alleles of the two proteins, demonstrating their redundant nature, the requirement for maternal protein and the fact that *Sox2* is more sensitive to reduced dosage than is Cdx2. They then went on to ask whether Lats kinase acts through Yap to mediate cell position during the cleavage stages leading up to blastocyst formation. They claim that overexpressing Lats2 reduces expression of PARD6B and aPKCz in cells and leads to their internalization, alongside a reduction in the number of outside versus inside cells.

Overall this is a carefully carried out study that provides some more insights into the mechanisms of the first lineage segregation in the mouse embryo. The strength of the paper is the description of the allelic series of mutations in *Yap/Wwtr1*, which have not been previously described. All previous studies depended largely on dominant negative and overexpression studies, which have their limitations. What is less strong is the section proposing a direct role for Lats in regulating polarity via downregulation of PARD6B and aPKCz. This underlies their model that Hippo signaling interaction with polarity components acts as a failsafe feedback mechanism to ensure lineage segregation. As outlined below, the data as presented raise some issues that need further resolution. In addition, there is no clear molecular mechanism proposed by which Lats activity would regulate specifically PARD6B and aPKCz and not other polarity components including the actin domain and phospho-ERM. The linkage between Lats, the apical domain, ROCK, etc. is presented in a model but there are many missing links in the model and a failure to link to other models of how polarity is thought to control Hippo signaling.

1) The first section on ROCK1/2 being upstream regulators of Cdx2 and *Sox2* expression is not well connected to the rest of the paper. They also did not actually look at the effects of Rocki on both Cdx2 and *Sox2* in the same embryos. Are the outside cells that remain *Sox2*-ve expressing Cdx2? Inhibition of Rock has multiple effects on the cell- which downstream response do they consider to be the critical one? Are they proposing a direct effect on Lats- as shown in Figure 7? Or the more usually suggested effect on the cortical actin domain, thus disrupting the segregation of Lats2? Or an effect on aPKC? All might be possible, but do they have any evidence for one versus the other?

2) Much of the conclusion on the involvement of *Yap/WWtr1* and Lats on changing the behavior of cells depends on the scoring of cells as inside or outside. The authors define inside cells as "appearing inside and showing uniform CDH1 over the cell surface". It is not clear exactly how these criteria were applied. What does it mean to 'appear' internal? 3D reconstructed from z stacks or just estimated from midline optical sections? I am not sure that this is a very accurate way to determine inner and outer cells and indeed they note that uniform CDH1 was not always a good predictor of position. This is a key point, because they then go on to claim that loss of *Yap/WWTr1* or activation of Lats leads not just to internalization of Lats-overexpressing cells, but to a shift in the actual proportion of inside and outside cells, with loss of outside and gain of inside cells. It is hard to understand how this can occur topologically, if the total cell number is unchanged, because reduction in outside cells would presumably lead to bulging of inside cells to the outside. Or are they proposing that the reduced number of outside cells somehow stretch out over the enlarged group of inner cells? The resolution of the images provided does not really resolve this issue.

3) It is not clear to me how the apparent reduction in expression of PARD6B and aPKCz in Lats-overexpressing cells is proposed to alter polarity and contractility, leading to internalization of the cells, given that the cells actually remain polarized as judged by other markers. Are they proposing a specific phosphorylation event that would alter cell polarity and contractility? Recent work from other labs has suggested that differential contractility is key to internalization of blastomeres during cleavage (e.g. Maitre et al., 2016)- have they looked at actomyosin? What happens to other components of the Hippo signaling pathway?

4) The model proposed in Figure 7 has ROCK as the factor linking the cell membrane to Lats regulation, but no specific mechanism is proposed. This model does not include the data from the Sasaki lab suggesting that the apical actin domain in the outside cells binds Lats and segregates it from its active complex with Nf2/Angiomotin/E-cadherin, thus reducing Hippo signaling in outside cells. ROCK could be involved in regulating the cortical actin domain, but has several other roles in the cell. A more comprehensive model, including data from other groups should be developed.

*Reviewer #3:*

The manuscript by Frum and Ralston reports an analysis of the role of the Hippo signalling pathway during the differentiation between trophectoderm and ICM.

They first show that the exclusion of SOX2, an inner cell marker, from outer cells depends on the activity of Rock as the use of a pharmacological inhibitor causes SOX2 expression in outer cells. The ectopic expression of a constitutively activated YAP prevents SOX2 expression in inner cells, showing nuclear Yap inhibition on SOX2 expression. This data reinforces their previous findings using Tead4 loss of function.

They then analysed the loss of function of *Yap1* and *Wwtr* genes in compound mutants. Their analysis shows that the phenotypes are linked to the number of alleles present, in a dose dependent manner, and that maternal expression partially rescues the loss of zygotic expression. The lower the dose of "Yap/Wwtr", the less CDX2 expression in outer cells and the more SOX2 expression in outer cells. They nicely show that completely removing the alleles by maternal-zygotic double deletion fully abolishes CDX2 expression and causes SOX2 expression in all the cells. Moreover, the loss of "Yap/Wwtr" is also correlated with a lower number of outside cells. Gain of function experiments with ectopic expression of Lats not only induced ectopic SOX2 expression but also decreases the number of outer cells and increases the number of inside cells. This lead them to the conclusion that ectopic Lats expression induces inside cell repositioning. Ectopic Lats expression phenotype can be partially rescued by co-expressing constitutively activated YAP. The mutation of *Sox2* does not interfere with YAP expression and activity, showing that SOX2 is only a marker (an important one) but not an actor. Finally, using Lats ectopic expression in outside cells, they show that the Hippo pathway can strikingly downregulate some polarity markers (Pard6B/aPKCz) but maintain others (Cdh1, pERM).

Their results are clearly described. They report an important analysis of Yap;Wwtr compound mutants, an involvement of the Hippo pathway for cell (re-)positioning and an effect of ectopically activated Hippo in outside cells on cell polarisation.

I suggest to address the following points:

1) The number of mutants is very low even if the phenotype seems to be fully penetrant. Would it be possible to increase the numbers, at least for the MZ-double mutants (2 is very/too small).

2) What is the proportion of outside cell death in these mutant embryos?

3) It is proposed that caYAP can rescue cell survival in Lats overexpressing cells. Can cell counts confirm this (only proportions between inside and outside are given).

[Editors’ note: what now follows is the decision letter after the authors submitted for further consideration.]

Thank you for resubmitting your work entitled "HIPPO signaling resolves embryonic cell fate conflicts during establishment of pluripotency in vivo" for further consideration at *eLife*. Your revised article has been favorably evaluated by Marianne Bronner (Senior Editor), a Reviewing Editor, and two reviewers.

The manuscript has been improved but there are some remaining issues that need to be addressed before acceptance, as outlined below.

We would recommend that you remove the experiment involving the LATS2-kinase dead construct, since the data are difficult to interpret. Removing the work will not impact on the primary conclusions of the paper, and indeed was not requested by either of the reviewers.

*Reviewer #1:*

The authors have carefully responded to the critiques and addressed most of the issues raised. The paper adds new information on the complexity of lineage segregation in the early embryo that will be of value to the field and also opens up new questions for investigation.

*Reviewer #2:*

With this revised version, the manuscript has appreciably improved with more numbers for the mutants (and more analyses) and better explanations in the analyses.

My only negative comment will concern the LATS2-kinase dead experiments that do not seem to work as YAP expression is not nuclear in inside GFP expressing cells (at least on the picture shown in Figure 6—figure supplement 1). This could be an explanation why it does not alter cell position. An appropriate embryo/section showing YAP nuclear expression in inside cells should be presented to allow concluding on the experiment.

Concerning the final interpretation, do the authors think that YAP and WWTR1 inhibit apoptosis? Could apoptosis be the result of an unresolved conflict between inside and outside (as they are still partially polarized), whereas cells moving in may have resolved the conflict by becoming inside cells?

---

## [Author Response]

[Editors’ note: the author responses to the first round of peer review follow.]

[…] As you will see both reviewers were very positive and feel that your genetic approach certainly has provided further insights into lineage segregation in the early embryo. However the biggest issue seems to be the study is a bit premature given the small numbers of embryos you've been able to analyse to date. If you can considerably improve the manuscript along the lines suggested by the reviewers, including expanding the samples sizes for critical genotypes we would be happy to see a major revision of the paper in the coming months, which would be considered as a new submission.

I am pleased to communicate that we have addressed all of the reviewers’ questions and concerns with new data. Our major changes include:

• Increased sample size for the critical genotypes, as requested.

• Additional phenotypic characterizations requested, including cell death assays.

We would like to take the opportunity to explain the novelty and significance of our study.

• No other study has examined the consequences of the combined loss of maternal and zygotic *Yap1* and *Wwtr1* in the mouse.

• We report new and unexpected roles for *Yap1/Wwtr1* in repressing expression of *Sox2*, the earliest known marker of pluripotency in the embryo.

• We report new and unexpected roles for *Yap1/Wwtr1* in promoting outside cell positioning.

• We identify the mechanism by which YAP1/WWTR1 promotes outside cell positioning by promoting formation of the apical domain – a process not previously defined.

Reviewer #2:[…] Overall this is a carefully carried out study that provides some more insights into the mechanisms of the first lineage segregation in the mouse embryo. The strength of the paper is the description of the allelic series of mutations in Yap/Wwtr1, which have not been previously described. All previous studies depended largely on dominant negative and overexpression studies, which have their limitations. What is less strong is the section proposing a direct role for Lats in regulating polarity via downregulation of PARD6B and aPKCz. This underlies their model that Hippo signaling interaction with polarity components acts as a failsafe feedback mechanism to ensure lineage segregation. As outlined below, the data as presented raise some issues that need further resolution. In addition, there is no clear molecular mechanism proposed by which Lats activity would regulate specifically PARD6B and aPKCz and not other polarity components including the actin domain and phospho-ERM. The linkage between Lats, the apical domain, ROCK, etc. is presented in a model but there are many missing links in the model and a failure to link to other models of how polarity is thought to control Hippo signaling.

We thank the reviewer for the very careful and thoughtful review. We found all of the reviewer’s questions and suggestions to be extremely valuable. Accordingly, we have performed the requested experiments and analyses to increase sample sizes and solidify the mechanism. These are detailed below.

1) The first section on ROCK1/2 being upstream regulators of Cdx2 and Sox2 expression is not well connected to the rest of the paper. They also did not actually look at the effects of Rocki on both Cdx2 and Sox2 in the same embryos. Are the outside cells that remain Sox2-ve expressing Cdx2?

We have performed the requested analysis of CDX2 and *SOX2* in ROCKi-treated embryos (Figure 1E and Figure 1—figure supplement 1A). In short, many outside cells in ROCKi-treated embryos coexpress *SOX2* and CDX2. We have revised the text to better connect these observations to the rest of the paper and address the reviewer’s questions in the last paragraph of the subsection “Patterning of *Sox2* is ROCK-dependent”.

Inhibition of Rock has multiple effects on the cell- which downstream response do they consider to be the critical one? Are they proposing a direct effect on Lats- as shown in Figure 7? Or the more usually suggested effect on the cortical actin domain, thus disrupting the segregation of Lats2? Or an effect on aPKC? All might be possible, but do they have any evidence for one versus the other?

We do not yet know the biochemical mechanisms by which ROCK influences *Sox2* expression. Since ROCKi has been shown to alter YAP1 localization (Kono et al., 2014), we propose that at least part of the ROCKi effect on *Sox2* is through its role in antagonizing YAP1 and WWTR1 activity, since ROCKi phenocopies *Wwtr1* and *Yap1* loss of function.

We now address this comment in the revised Discussion: “Here, we identify YAP1/WWTR1 as key components that repress *Sox2* expression in outside cells of the embryo. Notably, manipulations known to antagonize YAP1/WWTR1 activity, including chemical inhibition of ROCK and overexpression of LATS2 lead to ectopic expression of *SOX2* in outside cells, reinforcing the notion that YAP1/WWTR1 activity are crucial for repression of *Sox2* in outside cells.”

2) Much of the conclusion on the involvement of Yap/WWtr1 and Lats on changing the behavior of cells depends on the scoring of cells as inside or outside. the authors define inside cells as "appearing inside and showing uniform CDH1 over the cell surface". It is not clear exactly how these criteria were applied. What does it mean to 'appear' internal? 3D reconstructed from z stacks or just estimated from midline optical sections? I am not sure that this is a very accurate way to determine inner and outer cells and indeed they note that uniform CDH1 was not always a good predictor of position.

Indeed, CDH1 localization was not usually our method for determining cell position. We thank the reviewer for inviting the opportunity to clarify the methods used throughout the paper, we have added a section to the Materials and methods section entitled “Embryo Analysis.” Briefly, cell position was determined based on whether each individual cell in each section of each embryo made contact with the embryo’s external environment (outside) or whether it was surrounded by other cells (inside).

This is a key point, because they then go on to claim that loss of Yap/WWTr1 or activation of Lats leads not just to internalization of Lats-overexpressing cells, but to a shift in the actual proportion of inside and outside cells, with loss of outside and gain of inside cells. It is hard to understand how this can occur topologically, if the total cell number is unchanged, because reduction in outside cells would presumably lead to bulging of inside cells to the outside. Or are they proposing that the reduced number of outside cells somehow stretch out over the enlarged group of inner cells? The resolution of the images provided does not really resolve this issue.

The fewer outside cells appear to spread over the inner cells in these mutants. We have provided additional images and quantification of this phenotype in Figure 6, which we hope better illustrate this unusual phenotype. In addition, we describe the phenotype: “Critically, the fewer outside cells apparent in embryos lacking *Wwtr1* and *Yap1*, which appeared stretched over the mass of inside cells, exhibited ectopic expression of *SOX2* (Figure 6E-F).”

3) It is not clear to me how the apparent reduction in expression of PARD6B and aPKCz in Lats-overexpressing cells is proposed to alter polarity and contractility, leading to internalization of the cells, given that the cells actually remain polarized as judged by other markers. Are they proposing a specific phosphorylation event that would alter cell polarity and contractility? Recent work from other labs has suggested that differential contractility is key to internalization of blastomeres during cleavage (e.g. Maitre et al., 2016)- have they looked at actomyosin? What happens to other components of the Hippo signaling pathway?

We now discuss the reviewer’s questions: “[…] since PARD6B/aPKC are essential for outside cell positioning (Dard et al., 2009; Hirate et al., 2015; Plusa et al., 2005), the loss of the apical domain could affect cell positioning in several ways. For instance, loss of PARD6B/aPKC would eventually lead to cell depolarization (Alarcon, 2010), which could influence any of the processes normally governing the formation of inside cells, such as oriented cleavage, cell contractility, or apical constriction (Korotkevich et al., 2017; Maître et al., 2016; Samarage et al., 2015).”

Our lab intends to evaluate these mechanisms (including actomyosin and other HIPPO pathway members) in our future investigations.

4) The model proposed in Figure 7 has ROCK as the factor linking the cell membrane to Lats regulation, but no specific mechanism is proposed. This model does not include the data from the Sasaki lab suggesting that the apical actin domain in the outside cells binds Lats and segregates it from its active complex with Nf2/Angiomotin/E-cadherin, thus reducing Hippo signaling in outside cells. ROCK could be involved in regulating the cortical actin domain, but has several other roles in the cell. A more comprehensive model, including data from other groups should be developed.

We are grateful for the recommendation to develop a more inclusive model (Figure 7).

Reviewer #3:[…] I suggest to address the following points:1) The number of mutants is very low even if the phenotype seems to be fully penetrant. Would it be possible to increase the numbers, at least for the MZ-double mutants (2 is very/too small).

We have increased sample sizes as requested – the phenotypes are fully penetrant.

2) What is the proportion of outside cell death in these mutant embryos?

We have now evaluated cell death by TUNEL assay both Lats2-overexpressing (Figure 2—figure supplement 1) and *Wwtr1/Yap1* double knockout embryos (Figure 6G and Figure 6—figure supplement 1A). As anticipated, we observe increased cell death in both genotypes.

3) It is proposed that caYAP can rescue cell survival in Lats overexpressing cells. Can cell counts confirm this (only proportions between inside and outside are given).

We have also addressed this question by TUNEL assay (Figure 2—figure supplement 1). We observe decreased cell death in these embryos compared with embryos overexpressing *Lats2* alone, consistent with our proposal that caYAP can rescue cell survival.

[Editors' note: the author responses to the re-review follow.]

The manuscript has been improved but there are some remaining issues that need to be addressed before acceptance, as outlined below:We would recommend that you remove the experiment involving the LATS2-kinase dead construct, since the data are difficult to interpret. Removing the work will not impact on the primary conclusions of the paper, and indeed was not requested by either of the reviewers.Reviewer #2:With this revised version, the manuscript has appreciably improved with more numbers for the mutants (and more analyses) and better explanations in the analyses.My only negative comment will concern the LATS2-kinase dead experiments that do not seem to work as YAP expression is not nuclear in inside GFP expressing cells (at least on the picture shown in Figure 6—figure supplement 1). This could be an explanation why it does not alter cell position. An appropriate embryo/section showing YAP nuclear expression in inside cells should be presented to allow concluding on the experiment.

We have made the suggested changes and removed the LATS2-kinase dead experiments that the reviewers found to be ambiguous.